# Corneal Analysis with Swept Source Optical Coherence Tomography in Patients with Coexisting Cataract and Fuchs Endothelial Corneal Dystrophy

**DOI:** 10.3390/diagnostics11020223

**Published:** 2021-02-02

**Authors:** Anna Nowińska, Edyta Chlasta-Twardzik, Michał Dembski, Klaudia Ulfik-Dembska, Edward Wylęgała

**Affiliations:** 1Chair and Clinical Department of Ophthalmology, Faculty of Medical Sciences in Zabrze, Medical University of Silesia, 40-055 Katowice, Poland; edyta.chlasta@gmail.com (E.C.-T.); michau32@gmail.com (M.D.); klaudia.ulfik@gmail.com (K.U.-D.); rekroz@sum.edu.pl (E.W.); 2Ophthalmology Department, Railway Hospital, 40-760 Katowice, Poland

**Keywords:** fuchs endothelial corneal dystrophy, optical coherence tomography, cataract, corneal parameters

## Abstract

This study focused on defining the characteristic features of keratometry and pachymetry elevation maps based on swept source optical coherence tomography (SS OCT) in Fuchs endothelial corneal dystrophy (FECD) eyes with a coexisting cataract. 70 eyes of 35 patients diagnosed with FECD and a coexisting cataract and 70 control eyes were included in this prospective, controlled, observational, cross-sectional study. Features characteristic of intermediately affected eyes included an increased corneal thinnest thickness (CTT) (*p* = 0.01), 3 and 6 mm asymmetry (*p* < 0.0001), higher order Fourier indices (*p* < 0.05 and *p* ≤ 0.0001, respectively), chord µ, and a posterior Ectasia Screening Index (pESI) (*p* < 0.01). The lack of agreement between the anterior and posterior elevation map and a significant area of negative values in the posterior map were detected. In advanced FECD eyes, our study additionally revealed decreased posterior keratometry steep (Ks), keratometry flat (Kf), keratometry average (AvgK), eccentricity (Ecc), an increased corneal apex thickness (CAT), and decreased 3 and 6 mm posterior spherical indices (*p* < 0.0001 for all of the above). Characteristic features of subclinical FECD, independent of the corneal thickness, can be detected by SS OCT and should be considered during the preoperative assessment of patients with a coexisting cataract.

## 1. Introduction

Cataracts, according to World Health Organization (WHO) data, remain one of the main leading causes of vision impairment globally [1]. The National Eye Institute has established that the risk of cataracts increases with each decade of life, starting around the age of 40. Phacoemulsification with intraocular lens implantation surgery has been proven to be safe, with very few surgical complications, and very effective in terms of the predictability of refraction and visual outcomes [2,3]. However, optimal precision in the refractive outcome in cataract surgery depends on a precise preoperative assessment and attention to pre-existing ocular comorbidities [2,3,4]. The proportion of coexisting ocular comorbidities according to the European Registry of Quality Outcomes for Cataract and Refractive Surgery (EUREQUO) is high and ranges from 27 to 30% and should be taken into consideration while qualifying patients for cataract surgery [2].

Fuchs endothelial corneal dystrophy (FECD) is the most common primary corneal endothelial dystrophy and one of the leading indications for corneal transplantation worldwide [5,6,7]. FECD is a complex and heterogeneous genetic disease, and is characterized by progressive endothelial morphology changes, a decreased density of corneal endothelial cells, the formation of extracellular matrix excrescences called guttae and subsequent pathological changes to the corneal stroma, corneal edema, subepithelial fibrosis, vascularization, and corneal epithelium disruption [5,6]. FECD typically manifests in the fifth or sixth decades of life, with a greater incidence in women (female:male ratio of 3–4:1) [6]. The estimated prevalence of FECD varies among populations. In Caucasians, cornea guttata was diagnosed in 11% of females and 7% of males of the population over the age of 55 [8]. The cumulative incidence of cornea guttata for the age interval of 55–84 years in at least one eye, genders combined, was estimated to be 6.5–11.9% [9]. In the United States, the prevalence of FECD was estimated to be approximately 3.9–6.62% of the population [10,11].

Due to an ageing population and an increase in the life expectancy, the probability of diagnosing a patient with FECD and a coexisting certain degree of cataract is high. In these cases, visual disturbances can result from the FECD, the cataract, or both, offering a challenge for the surgical decision-making process [12]. Depending on the FECD severity grade, the management of such patients may include cataract surgery alone, or in combination with posterior lamellar or penetrating keratoplasty. One also has to take into a consideration a cataract surgery-induced endothelial cell loss of approximately 10%. A detailed preoperative corneal assessment is crucial to achieving the best postoperative results after cataract or transplantation surgery in FECD patients [13,14].

Several methods have been proposed for identifying patients at risk of the progression of FECD and corneal decompensation post-cataract surgery. Such predictors of disease progression include the central corneal thickness; endothelial cell density; corneal central-to-peripheral thickness ratio (CPTR), which is based on the ratio of the central corneal thickness and the peripheral corneal thickness at 4 mm from the center; backscatter of the basal epithelial cells using confocal microscopy or corneal backscatter (densitometry) using Scheimpflug imaging; 3D corneal shape deformation maps based on the Orbscan II anterior segment analysis system; 3D thickness maps generated from high-definition optical coherence tomography; and assessment of pachymetry map and posterior elevation map patterns also derived from Scheimpflug tomography [13,15,16,17,18,19,20,21,22,23].

Until now, the detailed corneal characteristics in patients with coexisting FECD and cataracts using swept source optical coherence tomography (SS OCT) have not been published. Currently, several OCT techniques are available, which differ in terms of the resolution of the obtained images, duration of the examination, range of the analyzed parameters (posterior and anterior segment of the eye), and possibility of performing morphometric measurements. There are three main OCT techniques: Time domain OCT; spectral or Fourier domain OCT (FD OCT); and OCT using swept source lasers (SS OCT). SS OCT is a variation of Fourier domain OCT, which detects the interference signal by using a wavelength scanning laser source of 1310 nm and a point photo detector [24]. The commercially available tomograph OCT Casia2 (Tomey, Japan) was introduced in 2008 and is characterized by a scanning speed of 50,000 A-scans/s, longitudinal resolution of 10 μm, and transverse resolution of 30 μm. The scan frame size is 16 × 13 mm, thus allowing data to be obtained from the corneal surface to the posterior surface of the lens in a single frame. Its rapid scan frequency and ability to scan the entire anterior segment from the anterior corneal surface to posterior lens surface within 0.3–2.4 s minimize motion artefacts. Consequently, the advantage of SS OCT is its higher robustness against sample motions than FD OCT, because of its short exposure time and imaging mechanism allowing data acquisition with a greater sensitivity and depth. Furthermore, 1310 nm penetrates opaque tissues more deeply, providing clearer images of cloudy corneas than visible wavelengths [25]. The SS OCT is characterized by a high repeatability and reproducibility of the keratometry, pachymetry, Fourier, corneal elevation, anterior eye chamber, and lens parameter results, regardless of the accommodative stress, which has been confirmed in previous studies [26,27,28].

This study was designed to identify differences between anterior and posterior corneal surface parameters, as well as Fourier indices, elevation maps, and the corneal thickness in patients with a coexisting cataract and Fuchs endothelial corneal dystrophy, which could have significant potential value in preoperative assessment for phacoemulsification and corneal transplant surgery. The study addresses the need to define SS OCT corneal morphometry features at different FECD severity grades, which could be responsible for postsurgical suboptimal visual and refractive results of patients with FECD.

## 2. Materials and Methods

The prospective, controlled, observational, cross-sectional study was conducted in accordance with the ethical standards stated in the 1964 Declaration of Helsinki and was approved by the Ethics Committee of the Medical University of Silesia, Katowice, Poland (KNE/0022/KB1/43/I/14; 1 July 2014). All patients had to sign an informed consent form before any study procedure. In total, 70 eyes of 35 patients diagnosed with FECD and a coexisting cataract (study group) and 70 eyes of 35 patients with a diagnosed cataract (control group) were included in the study.

The inclusion criteria for the study group included the clinical diagnosis of Fuchs endothelial corneal dystrophy and a coexisting cataract. The exclusion criteria were the presence of other ophthalmic or systemic diseases affecting the corneal morphology, and systemic or topical medication use known to affect the cornea. Eyes of patients who underwent ocular surgeries and eyes with refractive error higher than or equal to ±3.0 D were also excluded. The mean age of the patients was 64.72 ± 5.73 years (range of 50–76); there were 23 women and 12 men. The inclusion criteria for the control group were as follows: Cataract diagnosis; refractive error less than or equal to ±3.0 D; and no history of other ocular disease or surgery. The mean age of the subjects was 62.45 ± 6.89 years (range of 46–74); there were 22 women and 13 men. The control group subjects’ age and sex were matched with the study group. Demographic data are summarized in Table 1.

Clinical examination consisted of visual acuity, slit-lamp biomicroscopy with photography (SL 9900, Haag-Streit type, CSO, Scandicci, Italy), and anterior eye segment spectral swept source optical coherence tomography (SS OCT; OCT CASIA2; Tomey, Nagoya, Japan). The cataract severity and type were determined in the dilated eye with 0.5% tropicamide and 10% phenylephrine hydrochloride 30 min before examination according to the Lens Opacities Classification System III (LOCS III) [29] using a slit lamp at maximum illumination without light filtering. Six degrees of the extent of nuclear opacity (NO1–NO6) and nuclear color (NC1–NC6) were used as the parameters to determine the cataract density. Cortical (C1–C5) and posterior subcapsular (P1–P5) cataracts were graded using retroillumination to evaluate the degree to which the intrapupillary space or posterior capsule were obscured compared to standardized photographs. Grading of the FECD was accomplished by analyzing the cornea using the slit lamp biomicroscopy (10×, 16×, 25×, and 40× magnification), both with a narrow slit beam and using the retroillumination technique. The cornea was examined both horizontally and vertically from the limbus to the limbus, with special attention being paid to the central part and features characteristic of the disease, including corneal and stromal edema and clouding, a Descemet membrane of deep stromal folds, microcystic epithelial edema and bedewing, and subepithelial or intraepithelial bullae. FECD was assessed by using the modified reference Krachmer grading scale (7-step scale) [30]. The scale was as follows: Grade 0, no guttae; Grade 1, 1–12 central or paracentral non-confluent corneal guttae; Grade 2, more than 12 central and/or paracentral non-confluent corneal guttae; Grade 3, 1 to 2 mm of confluent central and/or paracentral corneal guttae; Grade 4, greater than 2 and up to 5 mm of confluent central and/or paracentral guttae; Grade 5, greater than 5 mm of confluent central and/or paracentral guttae; and Grade 6, over 5 mm of confluent central and/or paracentral guttae with clinically apparent stromal and/or epithelial edema. Using the updated grading scale, the eyes were further divided into three categories (Louttit, M.D. et al., 2012), consisting of unaffected, intermediately affected (grades 1–3), and affected (grades 4–6) groups [31]. Table 1 contains the study population characteristics.

Anterior segment imaging was performed by one observer (A. Nowińska). The examination was performed in a non-air conditioned, moderately lit room with no glare sources in the patient’s field of vision and the patient was instructed to blink twice before measurement. All examinations were performed between 10:00 and 13:00. During the SS OCT exam, we used the Corneal Map mode using 16 radial B-scans with 800 A scans per line sampling, the total scan duration was 0.3 s, the diameter was 16 mm, and the depth of the scan was 11 mm. The diameter employed for calculating the parameters of anterior and posterior corneal surfaces, as well as the corneal thickness and keratometry, was 10 mm. The quality check of the scans was performed in two steps: Automatically and manually. The SS OCT device instrument is equipped with an auto-alignment function and an auto-shot function that automatically initiated measurement when the subject’s eyes were within the proper range. Only measurements with a quality statement QS = ‘OK’ were accepted and further analyzed. QS is an index showing the reliability of the measured data for the following parameters: The offset degree of the corneal top is less than 0.86 mm (offset XY); area to be analyzed covers more than 93% of the front back corneal area of a 6 mm diameter (analyzed area); and contrast of the cornea is over 96.5% (valid data). When the measurement was completed, the preview screen with 16 radial OCT images with a green trace line on the corneal surfaces and crystalline lens was displayed and additionally, the correctness of the analysis was confirmed by two independent observers (A.Nowińska. and E.Chlasta-Twardzik). Images not accepted by any of the observers based on the alignment of the green trace line within the preview screen were excluded from the study.

A series of corneal shape and thickness parameters generated by the SS OCT (Tomey, Japan) software were studied: The keratometric (kKs, kKf; keratometric keratometry steep, flat), posterior (pKs, pKf; posterior keratometry steep, flat), and real (rKs, rKf; real keratometry steep, flat) steep and the flat meridian value of keratometry (D) and angle of axis (°); keratometric (kCYL, keratometric astigmatism), posterior (pCYL, posterior astigmatism), and real (rCYL, real astigmatism) astigmatism power (D); keratometric, posterior, and real average keratometry (D) (kAvgK, pAvgK, and rAvgK, respectively); eccentricity of the corneal curve (Ecc within 9.0 mm); area analyzed (AA, % within 10 mm); keratometric and real average central corneal power (kACCP and rACCP, respectively), anterior and posterior elevation at the 0, 180, 90, and 270 axis in four measurement locations, including central 0, 3, 5, and 7 mm (µm); keratometric and posterior 3 and 6 mm Fourier indices (spherical, regular, asymmetry, and higher-order); anterior and posterior ectasia screening index (aESI and pESI, respectively, %); corneal thickness on the measurement axis (µm); and thinnest corneal point thickness (value, location). The anterior chamber depth (ACD, mm) was also assessed. The X, Y coordinates (mm) of the pupil center were automatically measured. Based on the X, Y coordinates, the mean chord µ was established. Corneal morphology assessment was performed using 16 radial B-scans with 800 A scans per line sampling (11 × 16 mm). Characteristic corneal morphology features were described and compared for the study and control groups.

### 2.1. Corneal Shape Analysis

Corneal shape parameters were analyzed using the Corneal Map mode employing 16 radial OCT scans (800 A scans per line sampling). The refractive index of the Gullstrand model eye (cornea n_cornea_ = 1.376, aqueous humor n_aqueous humor_ = 1.336, and air n_air_ = 1.0) was used to calculate the map data. In addition, the Standard Keratometric Index (SKI) of 1.3375 was used analogous to a conventional topographic modeling system (TMS). The anterior, posterior, and real axial (instantaneous) power was calculated with appropriate formulas [4,20]. AvgK represents a mean value of steep meridian Ks and flat meridian Kf and was analyzed for the keratometric and posterior corneal surface. The real rAvgK, mean value of steep meridian Ks, and flat meridian Kf were based on the total power of anterior and posterior corneal surface. kACCP and rACCP represent the mean value of the keratometric and real axial power at a 3-mm diameter centered on the corneal apex, respectively. Ecc reflects the eccentricity of the quadratic curve within the 9.0 mm diameter area. AA represents the percentage of analysis range at a 10 mm diameter (assuming the whole circumference as 100%).

### 2.2. Corneal Thickness

Corneal Map mode using 16 radial OCT scans (800 A scans per line sampling) was applied to assess the corneal apex thickness (CAT) and corneal thickness of the thinnest point (CTT; corneal thinnest thickness). We used an auto-alignment function and an auto-shot function that automatically initiated measurement when the subject’s eyes were within the proper range. An offset degree of corneal top less than 0.86 mm (offset XY) was implemented by the producer software to qualify the image as accepted. The location of the thinnest corneal point in both horizontal and vertical planes based on the X, Y coordinates (mm) was also automatically measured for further analysis.

### 2.3. Elevation Map

The elevation maps of the anterior and posterior corneal surface were analyzed within a 9 mm diameter area. Color-coded maps with 20 µm steps were automatically implemented by the SS OCT software. In these maps, the green color represents a point on the Best Fit Sphere (BFS). The four measurement locations of central 0, 3, 5, and 7 mm at the 0, 180, 90, and 270 axis were used for further analysis.

### 2.4. Fourier Analysis and Ectasia Screening Index (ESI)

Fourier analysis was carried out automatically for the refractive power obtained from a keratometric and posterior topography map in the range of a 3 to 6 mm diameter. Analyzed parameters included the zero-order component (spherical component), first-order component (asymmetric component), second-order component (regular astigmatism component), and third or higher order component (higher order irregular astigmatism component). The Ectasia Screening Index (ESI) is a parameter employed for the detection of corneas with ectasia patterns. The analysis evaluates the anterior and posterior surfaces of the cornea to determine whether ectasia-specific patterns are present in a color map at a 6-mm diameter centered on the corneal apex. The quantitative index (%) was calculated based on the following points: Fourier analysis of the keratometric and posterior axial power data (Sph.6mm: spherical component (D) of Fourier analysis at a 6 mm diameter, Reg.6mm: regular astigmatism component (D) of Fourier analysis at a 6 mm diameter, Asy.6mm: asymmetric component (D) of Fourier analysis at a 6 mm diameter, and Hio.6mm: higher-order irregular astigmatism component (D) of Fourier analysis at a 6 mm diameter); the corneal thickness (μm) of the thinnest part and the location based on the apex of the cornea (X, Y coordinates) (mm); and the minimum value of the instantaneous posterior power and its location based on the apex of the cornea (X, Y coordinates) (mm) at a 6 mm diameter within an instantaneous posterior power map. The screening results were color-coded: Green for 0–4% (no pattern thought to be ectasia); yellow for 5–29% (patterns suspected to be ectasia); and red for 30% or higher (patterns that seem to be ectasia to some degree).

### 2.5. Statistical Analysis

Statistical analysis of the data obtained was carried out using the Statistica STatSoft program, version 13.1. (TIBCO Software Inc., Palo Alto, CA, USA) A p value of less than 0.05 was considered statistically significant. The mean values, standard deviation (SD), range, and 95% confidence interval (IC 95) were calculated for each parameter. Normal distribution was evaluated using the Shapiro–Wilk test. The analysis of the qualitative variables was conducted by calculating the number and percentage of the occurrence of each of the values. Differences between cohorts’ mean values were assessed using the Student *t* test. Whenever the parameters were not distributed normally, the Wilcoxon rank-sum test (Mann–Whitney *U* test) was used. To assess significant differences among more than two cohorts, the ANOVA Friedman test or Kendall’s W (Kendall’s coefficient of concordance) was used. The study group for statistical calculations was divided into three cohorts: Unaffected (70 eyes); intermediately affected (grades 1–3; 22 eyes); and affected (grades 4–6; 48 eyes). The intraclass correlation coefficient (ICC) was calculated to analyze the agreement between anterior and posterior elevation values and to assess mirror symmetry between left and right elevation maps. An ICC of 0 to 0.4 was considered weak; 0.4–0.7, moderate; 0.8 to 0.9, strong; and greater than 0.9, almost perfect.

## 3. Results

Table 1 contains the study population characteristics, including the range of FECD severity and LOCS value. The groups were not significantly different in terms of age, sex, and cataract severity grading.

### 3.1. Corneal Shape Analysis

We revealed a significant difference in the posterior corneal shape parameters of pKs, pKf, pAvgK, and the posterior eccentricity of the corneal curve within 9.0 mm (pEcc) (*p* = 0.0001, *p* < 0.0001, *p* < 0.0001, and *p* = 0.0001, respectively) between FECD and controls. Regarding the FECD severity grading, the difference in all analyzed corneal parameters was significant between unaffected and affected eyes, with the exception of the intermediately affected eyes. We also reveled a significant difference in chord µ values between FECD and controls (*p* < 0.01). The results of the corneal shape parameters and differences between groups are summarized in Table 2 and Table 3.

### 3.2. Elevation Map

The study revealed significant differences in posterior elevation maps in the central 3 mm diameter area. Central posterior elevation in FECD ranged from −4.0 to 12.0 µm, while in the controls, there were no negative values in the center (range 0–9 µm; mean 2.92 ± 2.33 µm). In FECD, contrary to the control group, there was a significant area of negative values in the posterior elevation map within a 3 mm diameter. Negative values in the 3 mm ranged from 5 to 32 µm (mean 15.06 ± 7.94 µm) in FECD vs. 2 to 13 µm (mean 8.01 ± 3.34 µm) in controls, and positive values ranged from 2 to 25 µm (mean 15.0 ± 5.75 µm) in FECD vs. 3 to 18 µm (mean 9.92 ± 3.95 µm), respectively. The differences were significant for positive and negative values in the 3 mm diameter area, with *p* = 0.01 and *p* < 0.01, respectively.

All eyes showed positive values, with no significant difference in anterior central elevation: range of 1–10 µm (mean 3.57 ± 2.54 µm) in FECD, and range of 1–4 µm (mean 2.73 ± 1.06 µm) in controls. Comparable to posterior elevation data, the range of negative and positive values in FECD eyes was significantly higher for the 3 mm diameter results: Positive and negative values were 3–14 µm (mean 7.52 ± 3.43 µm) and 3–13 µm (mean 7.56 ± 3.01 µm), respectively vs. 1–6 µm (mean 3.84 ± 2.08 µm) and 2–9 µm (mean 4.47 ± 2.11 µm), respectively, in controls. The differences were significant for positive and negative values in the 3 mm diameter area (*p* < 0.001 and *p* = 0.001, respectively). The ICC maps showed almost perfect agreement for anterior elevation and posterior elevation maps in controls (all ICCs ≥ 0.97), while in FECD eyes, ICCs ranged from 0.09 to 0.43. Additionally, there was a significant, visible interruption in the mirror right-left eye symmetry. Both anterior and posterior elevation maps in controls showed a typical pattern, with a slightly elevated isthmus from the temporal periphery to the neutral or positive central cornea (anterior: range 1–4 µm; mean 2.73 ± 1.06 µm, and posterior: range 0–9 µm; mean 2.92 ± 2.33 µm). This pattern was interrupted in posterior elevation maps of FECD eyes due to a focal corneal surface depression within the 3 mm diameter with a lack of the mirror right-left symmetry present in all 48 affected eyes (100%) and 16 intermediately affected eyes (72%), which is presented in Figure 1.

### 3.3. Fourier Analysis and ESI

Table 4 Analyzed Fourier corneal indices. Further comparison based on Fuchs endothelial corneal dystrophy (FECD) severity is shown in Table 5.

The difference in the following anterior Fourier parameters was significant between unaffected and affected eyes: 3 and 6 mm k asymmetry (both *p* < 0.0001) and k higher order (*p* < 0.001 and *p* < 0.0001, respectively). The above-mentioned anterior Fourier indices were increased in both intermediately affected (*p* < 0.05 for 3 mm k higher order and *p* < 0.0001 for 6 mm k higher order and k asymmetry) and affected (*p* < 0.0001 for all parameters) FECD eyes. Moreover, 3 and 6 mm posterior spherical indices were significantly less negative in affected eyes, with the exception of the intermediately affected eyes compared to controls (*p* < 0.0001). Posterior 3 and 6 mm asymmetry and higher order were significantly increased in both intermediately affected and affected eyes (*p* ≤ 0.0001 for all parameters). The relationship between Fourier indices and the central corneal thickness calculated using Pearson correlation with significance results revealed no significant correlation. All ICCs ranged from 0.04 to 0.43.

The quantitative Ectasia Screening Index differed significantly for the posterior corneal surface. The anterior ESI result was 0 for all control and intermediately affected eyes, while for affected eyes, the mean result was 2.81 ± 5.91%, but the difference was not significant. On the contrary, posterior ESI was significantly increased for both intermediately affected eyes (6.19 ± 5.17%; *p* < 0.01) and affected eyes (14.97 ± 8.91%; *p* < 0.001).

### 3.4. Corneal Thickness and Anterior Chamber Depth

The CAT in intermediately affected eyes did not differ from the control group eyes (mean difference 5.57 ± 19.69 µm; *p* = 0.39), but the difference was significant for affected eyes (mean difference 73.65 ± 42.61 µm; *p* < 0.0001). The corneal thickness of the thinnest point (CTT) was increased significantly in both severity grades. The mean difference in intermediately affected eyes was 14.34 ± 17.98 µm (*p* = 0.01) and 64.86 ± 31.79 µm (*p* < 0.0001) in affected eyes. In all normal controls, the thinnest corneal point was located in the temporal-inferior quadrant, while this location was present in only 19 (27.14%) of FECD eyes. FECD eyes were characterized by a random thinnest point location and the difference was significant (*p* < 0.001). When comparing the thinnest corneal point location according to FECD severity, the dislocation was present in 9 (40.1%) intermediately affected eyes and 42 (87.5%) affected eyes.

There were no significant differences in the ACD in studied groups. The results are summarized in Table 2 and Table 3.

### 3.5. Corneal Morphology

High resolution corneal scans revealed a hyperreflective line at the border of the posterior corneal surface with an uneven continuity and wavy appearance, with numerous, small hyperreflective dots corresponding to “guttae” on slit lamp examination. An increased reflectivity was noted within the lens due to cataract opacities. A representative image is presented in Figure 1c.

## 4. Discussion

FECD is a progressive, complex disease in terms of its influence on the corneal morphology and morphometric features. In this study, we focused on defining the detailed characteristic features of keratometry and pachymetry elevation maps, as well as Fourier indices, based on the SS OCT of normal and FECD eye patients with a coexisting cataract with different grades of disease severity.

In terms of the corneal shape parameters, our study confirmed the decreased posterior corneal power of steep and flat meridian pKs, pKf, and posterior average keratometry pAvgK. Similar findings were previously reported by Wacker et al. based on a Scheimpflug imaging analysis in the case of moderate and advanced FECD and by Brunette at al. based on the Orbscan system [16,20]. In our study group, the difference was significant in affected eyes (Krachmer grades 4–6), with the exception of intermediately affected eyes (Krachmer grades 1–3), thus confirming that posterior keratometry progresses in the course of the disease and becomes evident in advanced stages. This change in the posterior corneal power should not be neglected, especially when selecting IOLs for cataract surgery or planning astigmatism correction using toric IOLs or limbal relaxing incisions. Moreover, the significant increase of the chord µ value in eyes without clinically significant edema may be responsible for the increased probability of the presence of halos and glare after phacoemulsification surgery [32]. Devices that combine high resolution corneal measurements with biometry using total keratometry measurements with software that uses vectorial calculation or ray tracing, swept source OCT, and implementation of the new calculation formulas, such as the Barrett toric calculator and the Abulafia–Koch formula, should be considered for IOL calculation, in order to improve the prediction of the postoperative refractive outcome [33]. Another corneal shape parameter which differed significantly between FECD and controls was the posterior eccentricity (pEcc). In our study, the mean values of anterior and posterior Ecc in controls were 0.61 ± 0.08 and 0.56 ± 0.17, respectively, which was similar to other authors’ findings. The mean eccentricity reported from different geographical regions varies between 0.27 and 0.66, and this inter-study variation of results could be explained by differences in the characteristics of the studied populations or different measurement devices [34]. Our study group was age comparable to the control group, so the effect of age on the change in Ecc index was clinically insignificant. According to the results of the present study, all normal and FECD eyes were characterized by a prolate shape, but the posterior eccentricity was significantly decreased (towards decreased prolateness) in affected FECD eyes (normal: 0.56 ± 0.17; intermediately affected: 0.54 ± 0.13, *p* = 0.25; affected: 0.18 ± 0.42, *p* < 0.0001). As eccentricity describes the rate of corneal flattening from the central cornea to the periphery, the result obtained can be explained by the fact that the central posterior cornea in FECD is less negative compared to controls. Moreover, based on previous studies on the posterior surface of the cornea and the effect of Fuchs dystrophy on the corneal thickness, we know that the cornea swells posteriorly more than it does anteriorly and the central posterior cornea is flatter and more spherical than normal, resulting in a less negative power and the loss of normal posterior surface toricity, which perfectly explains the pEcc result obtained in our study [16,17,20].

Historically, an increased central corneal thickness was one of two major parameters linked to FECD, along with the “cornea guttata”. Progressive cases of corneal edema with corneal “crystal-like guttae” were described as early as 1916 by Koeppe and in 1920 by Kraupa [30]. However, there are many limitations in assessing the corneal apex thickness/central corneal thickness (CAT/CCT) as a function of the disease progression. Normal corneas without edema can also have a thickness of 600 µm or more, which is why CAT should be studied with regards to the corneal thickness before the disease onset [35]. Furthermore, there is significant daily variation of the CAT in patients with FECD. In the morning, after eye opening, CAT can be 31 to 58 μm thicker, which resolves within four hours [36]. A similar study showed a difference of 41.4 μm, accompanied by a myopic shift and glare [37]. We revealed that in affected eyes, the CAT was significantly thicker when compared to normal controls (*p* < 0.0001), but there was no significant difference between normal and intermediately affected eyes. At present, due to the development of lamellar transplantation, CAT is no longer a sufficient and reliable parameter for defining disease progression, as most patients undergo surgery earlier in the course of the disease at stages without clinically manifesting edema, before the development of irreversible ultrastructural fibrotic changes in the corneal stroma. This statement was also postulated by the authors of previous studies [17,20,21,22,23]. For this reason, a series of objective pachymetry parameters was proposed. The corneal central-to-peripheral thickness ratio (CPTR), which is based on the ratio of the central corneal thickness and the peripheral corneal thickness at 4 mm from the center, was introduced by Repp at al. and was proven to be repeatable and highly correlated with the clinical grade of FECD, and to result in excellent discrimination between FECD and normal eyes [17]. In our opinion, the corneal thinnest thickness (CTT) and the displacement of the thinnest point of the cornea are more reliable parameters compared to the CAT and should be used to distinguish corneas with subclinical edema from normal corneas. In our study, CTT was significantly thicker in early disease grades: 530 ± 18.19 µm in intermediately affected eyes compared to 517 ± 12.83 µm in control eyes (*p* = 0.01) and 582 ± 30.27 µm (*p* < 0.0001) in affected eyes. The displacement of the thinnest point location from the temporal-inferior quadrant was also underlined in recent studies based on the Scheimpflug imaging, as one of the important predictors of disease progression [21,22].

Other studies based on Fourier domain OCT have focused on visualization and assessment of the three-dimensional (3D) endothelium and Descemet’s membrane complex thickness (En-DMT) [15,23,38]. A recent OCT study revealed that 3D En-DMT has the highest sensitivity and specificity in diagnosing and grading FECD compared to the regional total corneal thickness (TCT) and central-to-peripheral total corneal (CPTR). For identifying FECD, En-DMT parameters showed a specificity of 100% and sensitivity of 92% or more, while for discriminating between healthy and early-stage disease, En-DMT parameters exhibited a specificity of more than 92% with a sensitivity of 92%. The authors used an automated custom-built segmental tomography algorithm to segment the corneal boundaries and generate 3D thickness maps using an FD OCT system with the axial resolution of 3 µm [23]. The anterior segment SS OCT used in our analysis has a longitudinal resolution of 10 µm and has no automatic option to isolate the Descemet membrane within a scan frame. Our goal was to create a study that could produce results that are directly useful in everyday clinical practice. Therefore, we focused on corneal parameters available in CASIA2 software, which could be easily reproduced in clinical settings.

The morphological changes revealed by our study were also limited due to the relatively lower resolution (10 µm) than in other HD OCT devices (3 µm). To precisely assess the corneal morphology with the relation to corneal layers at initial stages of FECD, it would be advisable to use ultra-high resolution or 3D anterior segment OCT, used by Shousha et al. [15], Eleiva et al. [23], or Iovino at al. [38] in their research, or to use confocal microscopy characterized by an axial resolution of 1 µm. The other reason for this is that, nowadays, we do not observe advanced, irreversible corneal changes in the course of the FECD in a clinical setting, because patients qualify for surgeries earlier in the disease course. In our study group, the patient with the highest corneal thickness had CAT of 648 µm and CTT of 624 µm. Observations such as epithelial bullae, thickening of the epithelium, and subepithelial fibrosis, observed by Kaluzny et al. in 2009 in advanced FECD, were not present in our study group [39].

Fourier indices analysis is available in SS OCT and carried out automatically based on the refractive power obtained from a keratometric and posterior topography map in the range of a 3 and 6 mm diameter. The following indices, revealed by our study, could be of potential use for discrimination between intermediately affected eyes and controls: 3 and 6 mm k asymmetry (both *p* < 0.0001) and k higher order (*p* < 0.05 and *p* < 0.0001, respectively), and 3 and 6 mm p asymmetry and p higher order (*p* ≤ 0.0001). Moreover, the results of the anterior and posterior indices were independent from CAT, which proves that abnormalities appear early in the course of the disease. Wacker et al., in a study focused on analyzing wavefront errors over a 6-mm diameter optical zone and corneal backscatter by using a rotating Scheimpflug camera, also postulated the presence of early surface changes on both anterior and posterior corneal surfaces [19]. The pathophysiological explanation of the early changes mentioned above could be related to the ultrastructural changes in the anterior cornea, such as subepithelial fibrosis and keratocyte depletion, which might affect the corneal architecture, and to the changes in the posterior cornea, such as “corneal guttae”, which create an irregular posterior surface. This could explain suboptimal refractive and visual results of lamellar keratoplasty [18,40,41]. It should also be taken into account that such subtle changes in the surface of the cornea are not specific, and the influence of other factors on their appearance, such as disturbance of the tear film, internal/indoor environmental factors, the time of day, or the presence of other ocular factors, cannot be ruled out. Tear film instability is a well-recognized factor affecting the corneal topography reading [42]. Although patients were always asked to blink before measurement, the influence of the tear film disturbances could have potentially interfered with the Fourier indices results. We also revealed patterns suspected to be ectasia on the posterior corneal surface in the area of 6 mm in patients with early and advanced stages of the disease. Posterior pESI was significantly increased in intermediately affected and affected eyes (6.19 ± 5.17%; *p* < 0.01 and 14.97 ± 8.91%; *p* < 0.001 respectively), while the result was 0% for all control eyes. Such correlations with possible ectasia patterns were also previously reported. The association between keratoconus and FECD has been reported in the literature in various case reports [6]. Both diseases have common genetic and environmental pathogenesis factors: Polymorphisms in FLAP endonuclease (g.61564299G>T and c.-441G>A) and LIG3 (g.29661G>A and g.29059C>T), which encode DNA ligase III, oxidative stress, the formation of reactive oxygen species (ROS), and mitochondrial dysfunction [6]. The abnormal pESI result could also be explained by the disturbances in Fourier indices results, as the index was calculated based on the 6 mm Fourier indices results, the corneal thickness of the thinnest part, and the minimum value of the instantaneous posterior power.

The elevation map characteristic of FECD showed a lack of agreement between anterior and posterior values (ICCs 0.09–0.43 in FECD vs. all ICCs ≥ 0.97 in controls), a significant area of negative values in the posterior elevation map, an increased range of positive and negative values in the anterior and posterior elevation map within a 3 mm diameter compared to controls (all *p* ≤ 0.01), and a lack of mirror right-left eye symmetry. The right-left mirror symmetry is one of the features of healthy corneas, observed in the process of optimizing the design of biosynthetic cornea [43]. In total, 72% of intermediately affected and 100% of affected eyes lacked the symmetry in our study, mainly due to the focal posterior corneal surface depression and displacement of the thinnest point location. These features of the elevation map were also established based on Scheimpflug topography studies by Sun, S.Y. et al. and Patel, S.V. et al. [21,22]. The authors reported a loss of parallel isopachs, displacement of the thinnest point of the cornea, and focal posterior corneal surface depression. Based on these findings, a new classification of FECD severity was proposed, dividing eyes into the following categories: Eyes with clinically definite edema; eyes with subclinical edema; and eyes without edema. Moreover, the authors evaluated the prognosis of 96 eyes of 56 subjects over 60-month (range, 45–72 months) follow-up period. In multivariate analyses, a loss of regular isopachs (Hazard Ratio, 11.57; *p* < 0.001) and displacement of the thinnest point (HR, 5.61; *p* = 0.02) were independent and clinically important risk factors for progression and surgical intervention. The authors suggested that tomographic analysis should be applied to FECD corneas without clinically definite corneal edema to assess whether subclinical edema is present.

To the best of our knowledge, our study is the first observational study assessing the wide range of corneal parameters based on SS OCT at one time point in eyes with FECD of different severities. We did not evaluate the prognosis of the study group, but this evaluation would be valuable for future studies. The study group was included for further analysis and qualification for cataract surgery or lamellar keratoplasty. Our results cannot be directly compared to the Scheimpflug topography, because of significant systematic differences between the measurements of these two devices.

Features characteristic of intermediately affected eyes based on SS OCT included the following: An increase in CTT; displacement of the thinnest point of the cornea; increased chord µ; increased 3 and 6 mm keratometric and posterior asymmetry and higher order Fourier indices; pESI result of the suspected ectasia pattern; a lack of agreement between the anterior and posterior elevation map; and a significant area of negative values in the posterior elevation map. In advanced FECD eyes, our study additionally revealed a decreased posterior corneal power of steep and flat meridian pKs, pKf, and posterior average keratometry pAvgK; pECC (towards decreased prolateness); increased CATl and decreased 3 and 6 mm posterior spherical indices.

The study is not without certain limitations. The limitations of this study include the relatively small sample size (three cohorts: unaffected (70 eyes), intermediately affected (grades 1–3; 22 eyes), and affected (grades 4–6; 48 eyes). A larger series is needed to further divide eyes into four categories (unaffected, mild, moderate, and advanced). We will make an effort to increase the study group size for further analysis. Additionally, the impact of factors, such as disturbance of the tear film, internal/indoor environmental factors, the time of day, or the presence of other ocular factors, cannot be ruled out. Although all examinations were performed between 10:00 and 13:00, the impact of the daily hour on FECD could be significant [36,37]. Moreover, the repeatability and reproducibility of SS OCT were already proved in healthy corneas, but there is a lack of such studies in larger samples of FECD eyes [26,27,28]. Variations in corneal Fourier indices or elevation might occur due to subtle misalignments caused by eye movements during examination.

In summary, our study revealed a number of characteristic corneal features in early and advanced stages of FECD. Such corneal characteristic of FECD eyes, independent of the presence of clinically significant corneal edema, may lead to unexpected postoperative visual outcomes after cataract and combined cataract and transplantation surgery. A detailed preoperative corneal assessment in terms of the risk of progression could improve outcomes by optimizing patient selection and the surgical approach and improvements in postoperative visual acuity achievement.

Finally, as proposed for detecting progression in diabetic retinopathy, age-related macular degeneration or subclinical keratoconus automatic learning systems with artificial intelligence techniques may allow the development of an objective set of parameters for accurately diagnosing the risk of prognosis in FECD. The analysis of identified corneal predictors, in sync with their evaluated cut-off points, may become a guideline for cataract surgeons during qualification of these complicated, complex cases.

## 5. Conclusions

FECD is characterized by a progressive, increasing impact on posterior Fourier indices and posterior elevation map. Increase of the central corneal thickness should be regarded as an advanced sign of FECD. Characteristic features of subclinical FECD independent of corneal thickness can be detected by SS OCT and should be considered during preoperative assessment in patients with coexisting cataract.

## Figures and Tables

**Figure 1 diagnostics-11-00223-f001:**
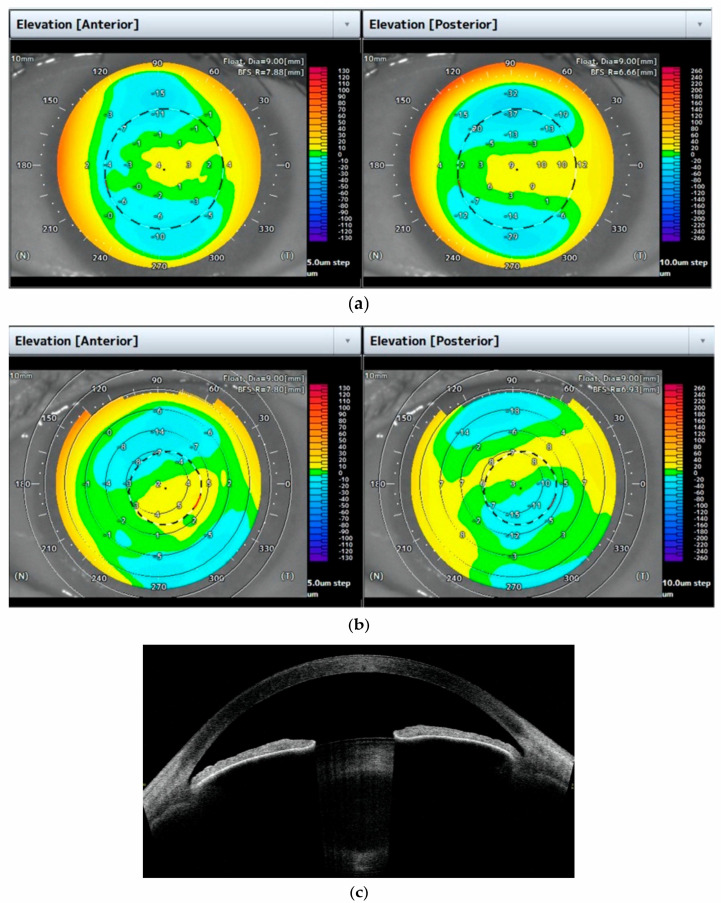
Anterior and posterior elevation maps of FECD and control eyes. Corneal and lens morphology assessment. (**a**) Maps of the left eye, representative of the control group. The maps show the typical characteristic pattern in which the apex is depicted in a warm colors above the Best Fit Sphere (BFS) with a slightly elevated isthmus which joined the central cornea from the temporal periphery. The maximum range between negative and positive values was within 15 and 27 µm in 3 and 5 mm diameters for anterior maps and within 31 and 69 µm in 3 and 5 mm diameters for posterior maps, respectively. The anterior and posterior maps showed almost perfect agreement (all intraclass correlation coefficients (ICCs) ≥ 0.97). (**b**) Maps of the left eye, representative of the FECD group (Grade 4, affected group). A significant area of negative values can be observed in the posterior elevation map within a 3 mm diameter exceeding 15 µm. The maximum range between negative and positive values in FECD was within 27 and 40 µm in 3 and 5 mm diameters for anterior maps and within 57 and 89 µm in 3 and 5 mm diameters for posterior maps, respectively. The values of the posterior elevation map do not correspond to the anterior elevation results (ICCs ranged from 0.22 to 0.53). (**c**) High resolution scan of the eye, representative of the FECD group (Grade 4, affected group). A hyperreflective thin line at the border of the posterior corneal surface with an uneven continuity and wavy appearance is noted. Areas of increased reflectivity can be observed within the lens due to cataract opacities.

**Table 1 diagnostics-11-00223-t001:** Study population characteristics. Significance level of *p* < 0.05 was marked as “*”.

Age, Sex, and Cataract Grading							
	Control				FECD		
Age mean ± sd/ median/ range	62.45 ± 6.89 63 46–74				64.72 ± 5.73 62 50–76		
Sex F/M (%)	23/12 (65.71/34.28)				22/13 (62.85/37.14)		
Cataract (LOCS III) mean ± sd	NO/NC 2.36 ± 0.78 C 2.15 ± 0.97 P 2.11 ± 1.14				NO/NC 2.28 ± 0.93 C 2.05 ± 0.86 P 2.01 ± 0.98		
**FECD Severity Grade**							
Krachmer scale [30]	grade 0	Grade 1 1–12 corneal guttae	Grade 2 >12 corneal guttae	Grade 3 1–2 mm diameter	Grade 4 2–5 mm diameter	Grade 5 >5 mm diameter	Grade 6 >5 mm diameter edema
Eyes	70	3	6	13	12	17	19
Louttit, M.D. et al. [31]	Unaffected	Intermediately Affected (grades 1–3)			Affected (grades 4–6)		
Eyes	70	22			48		
cBCVA mean ± sd range	0.5 ± 0.24 0.4–0.7	0.5 ± 0.31 0.3–0.8			0.3 ± 0.39 * 0.1–0.4		
Age mean ± sd/ median/ range	62.45 ± 6.89 63 46–74	63.98 ± 7.22 61 50–74			65.02 ± 6.94 63 53–76		

**Table 2 diagnostics-11-00223-t002:** Comparison of keratometry, pachymetry, and anterior chamber data of the Fuchs endothelial corneal dystrophy (FECD) eyes and control group.

Parameter	Controls (*n* = 70)		FECD (*n* = 70)		Mean Difference (IC 95)	*p*
	Mean ± sd	95%CI Min–Max	Mean ± sd	95%CI Min–Max		
kKs	44.0 ± 1.85	43.55–44.44 40.8–46.5	44.41 ± 0.82	44.21–44.61 42.4–45.4	−0.41 (−0.84 to 0.02)	0.06
kKf	43.0 ± 1.53	42.6–43.37 40.3–45.1	43.26 ± 0.73	43.08–43.43 41.7–44.5	−0.25 (−0.55 to 0.04)	0.09
kCYL	1.03 ± 0.7	0.86–1.19 0.4–2.6	1.13 ± 0.54	0.99–1.26 0.6–2.1	−0.09 (−0.13 to 0.11)	0.36
kAvgK	43.53 ± 1.67	43.13–43.93 40.6–45.4	43.78 ± 0.75	43.6–43.96 42.0–45.0	−0.25 (−0.58 to 0.08)	0.14
kEcc	0.61 ± 0.08	0.58–0.62 0.49–0.75	0.61 ± 0.17	0.56–0.64 0.32–0.97	−0.01 (−0.04 to 0.03)	0.84
kAA	96.03 ± 5.4	94.74–97.31 84.0–100.0	94.34 ± 4.81	93.19–95.48 89.3–100.0	1.69 (−0.07 to 3.45)	0.06
kACCP	43.51 ± 1.7	43.1–43.91 40.5–45.4	43.71 ± 0.73	43.53–43.88 42.0–44.9	−0.2 (−0.53 to 0.13)	0.23
pKs	−6.28 ± 0.31	−6.35 to −6.21 −6.6 to −5.9	−6.03 ± 0.24	−6.09 to −5.97 −6.3 to −5.7	−0.24 (−0.37 to −0.12)	**0.0001**
pKf	−5.97 ± 0.22	−6.03 to −5.92 −6.11 to −5.81	−5.73 ± 0.18	−5.77 to −5.68 −5.84 to −5.4	−0.24 (−0.31 to −0.17)	**<0.0001**
pCYL	0.31 ± 0.15	0.27–0.35 0.1–0.6	0.32 ± 0.16	0.26–0.34 0.1–0.7	0.01 (−0.01 to 0.02)	0.97
pAvgK	−6.12 ± 0.24	−6.17 to −6.06 −6.3 to −5.6	−5.86 ± 0.24	−5.92 to −5.8 −6.2 to −5.4	−0.25 (−0.36 to −0.15)	**<0.0001**
pEcc	0.56 ± 0.17	0.52–0.6 0.2–0.73	0.36 ± 0.42	0.26–0.47 0.06–0.85	0.19 (0.07–0.31)	**0.001**
pAA	95 ± 6.32	93.58–96.6 84–100.0	93.43 ± 5.33	92.16–94.71 86.1–100.0	1.57 (−0.26 to 3.57)	0.09
rKs	42.96 ± 1.84	42.52–43.41 39.9–45.3	42.48 ± 1.06	42.23–42.74 40.4–44.2	0.48 (0.05–0.9)	0.05
rKf	42.14 ± 1.57	41.77–42.52 39.5–44.2	42.45 ± 1.09	42.19–42.71 40.4–44.1	−0.31 (−0.57 to −0.04)	0.06
rCYL	0.93 ± 0.61	0.78–1.07 0.3–2.3	1.13 ± 0.55	1.00–1.26 0.4–2.2	−0.19 (−0.39 to −0.01)	0.05
rAvgK	42.86 ± 1.21	42.57–43.15 39.8–44.4	43.08 ± 0.91	42.86–43.3 40.9–44.8	−0.22 (−0.5 to 0.06)	0.12
rAA	94.55 ± 7.83	94.39–96.99 85.0–100.0	92.78 ± 6.54	90.98–94.02 82.3–100.0	1.77 (−0.16 to 3.21)	0.11
rACCP	42.41 ± 1.69	42.11–42.92 39.6–44.4	42.98 ± 0.96	42.75–43.21 40.6–44.6	−0.57 (−0.78 to −0.15)	0.06
CAT	536 ± 17.27	532.48–540.71 500.0–560.0	581 ± 45.68	570.36–592.09 510.0–648.0	45.0 (33.4–55.79)	**<0.0001**
CTT	517 ± 12.83	514.52–520.64 497.0–537.0	561 ± 36.82	553.01–570.58 505.0–624.0	44.01 (34.76–53.63)	**<0.0001**
ACD	3.14 ± 0.74	3.01–3.19 2.84–3.72	3.01 ± 0.83	2.97–3.43 2.75–3.41	0.13 (−0.12 to 0.54)	0.89
chord µ	0.35 ± 0.11	0.32–0.38 0.1–0.6	0.42 ± 0.1	0.4–0.44 0.2–0.6	0.07 (0.3–0.1)	**<0.01**

**Table 3 diagnostics-11-00223-t003:** Comparison of keratometry data of the intermediately affected, affected, and unaffected eyes.

		Study Group vs. Controls Keratometry, Pachymetry, and Anterior Chamber Data		
	Mean ± sd; (95% CI) *p*			
FECD Grade		Unaffected *n* = 70	Intermediately Affected (Grades 1–3) *n* = 22	Affected (Grades 4–6) *n* = 48
pKs		−6.28 ± 0.31 (−6.35 to −6.21)	−6.11 ± 0.16 (−6.19 to −6.04) *p* = 0.14	−5.96 ± 0.26 (−6.04 to −5.88) ***p* < 0.0001**
pKf		−5.97 ± 0.22 (−6.03 to −5.92)	−5.84 ± 0.18 (−5.93 to −5.76) *p* = 0.14	−5.66 ± 0.16 (−5.71 to −5.61) ***p* < 0.0001**
pAvgK		−6.12 ± 0.24 (−6.17 to −6.06)	−5.99 ± 0.13 (−6.05 to −5.93) *p* = 0.15	−5.77 ± 0.25 (−5.85 to −5.70) ***p* < 0.0001**
pEcc		0.56 ± 0.17 (0.52–0.6)	0.54 ± 0.13 (0.52–0.7) *p* = 0.25	0.18 ± 0.42 (0.06–0.31) ***p* < 0.0001**
CAT		536 ± 17.27 (532.48–540.71)	540.47 ± 20.25 (530.82–549.26) *p* = 0.39	608.52 ± 36.31 (597.83–619.21) ***p* < 0.0001**
CTT		517 ± 12.83 (514.52–520.64)	530 ± 18.19 (522.95–539.23) ***p* = 0.01**	582 ± 30.27 (373.41–590.99) ***p* < 0.0001**
chord µ		0.35 ± 0.11 (0.32–0.38)	0.42 ± 0.09 (0.39–0.44) ***p* < 0.01**	0.45 ± 0.1 (0.42–0.47) ***p* < 0.0001**

**Table 4 diagnostics-11-00223-t004:** Comparison of Fourier indices of the FECD eyes and control group.

Parameter	Controls (*n* = 70)		FECD (*n* = 70)		Mean Difference (IC 95)	*p*
	Mean ± sd	95%CI Min–Max	Mean ± sd	95%CI Min–Max		
3 mm k Spherical	43.79 ± 1.43	43.44–44.14 40.3–45.6	43.69 ± 0.79	43.42–43.98 42.0–44.69	0.09 (−0.31 to 0.49)	0.66
3 mm k Reg, Astigmatism	0.53 ± 0.26	0.46–0.59 0.21–1.32	0.62 ± 0.39	0.49–0.61 0.31–1.55	−0.09 (−0.18 to 0.03)	0.18
3 mm k Asymmetry	0.28 ± 0.14	0.24–0.31 0.09–0.58	0.69 ± 0.28	0.64–0.69 0.41–1.13	−0.41 (−0.48 to −0.34)	***p* < 0.0001**
3 mm k Higher Order	0.14 ± 0.02	0.13–0.15 0.09–0.21	0.19 ± 0.03	0.18–0.22 0.14–0.22	−0.05 (−0.06 to −0.04)	***p* < 0.001**
6 mm k Spherical	43.67 ± 1.37	43.34–43.99 40.41–45.5	43.52 ± 0.71	43.46–43.89 41.93–44.5	0.14 (−0.23 to 0.53)	0.45
6 mm k Reg, Astigmatism	0.48 ± 0.27	0.42–0.55 0.17–1.29	0.56 ± 0.36	0.43–0.81 0.27–1.46	−0.08 (−0.18 to 0.02)	0.13
6 mm k Asymmetry	0.36 ± 0.19	0.31–0.40 0.14–0.81	0.87 ± 0.29	0.83–1.13 0.62–1.58	−0.51 (−0.58 to −0.44)	***p* < 0.0001**
6 mm k Higher Order	0.15 ± 0.03	0.15–0.16 0.12–0.2	0.24 ± 0.04	0.21–0.27 0.13–0.45	−0.09 (−0.11 to −0.06)	***p* < 0.0001**
3 mm p Spherical	−6.15 ± 0.2	−6.19 to −6.10 −6.33 to −5.61	−5.86 ± 0.24	−5.89 to −5.69 −6.26 to −5.52	−0.29 (−0.36 to −0.21)	***p* < 0.0001**
3 mm p Reg, Astigmatism	0.15 ± 0.05	0.14–0.17 0.05–0.3	0.14 ± 0.09	0.13–0.17 0.07–0.41	0.01 (−0.02 to 0.02)	0.96
3 mm p Asymmetry	0.04 ± 0.02	0.03–0.05 0.01–0.1	0.28 ± 0.13	0.26–0.33 0.05–0.54	−0.24 (−0.27 to −0.21)	***p* < 0.0001**
3 mm p Higher Order	0.02 ± 0.01	0.01–0.02 0.01–0.06	0.05 ± 0.01	0.04–0.06 0.03–0.11	−0.03 (−0.03 to −0.02)	***p* < 0.0001**
6 mm p Spherical	−6.13 ± 0.19	−6.19 to −6.10 −6.36 to −5.92	−5.78 ± 0.31	−5.85 to −5.69 −6.3 to −5.56	0.35 (−0.38 to −0.24)	***p* < 0.0001**
6 mm p Reg, Astigmatism	0.16 ± 0.05	0.15–0.17 0.05–0.31	0.17 ± 0.12	0.11–0.24 0.08–0.51	−0.01 (−0.04 to 0.02)	0.47
6 mm p Asymmetry	0.05 ± 0.03	0.04–0.06 0.02–0.11	0.3 ± 0.13	0.32–0.38 0.06–0.56	−0.25 (−0.28 to −0.2)	***p* < 0.0001**
6 mm p Higher Order	0.03 ± 0.01	0.02–0.03 0.01–0.07	0.06 ± 0.02	0.05–0.07 0.03–0.12	−0.03 (−0.03 to −0.02)	***p* < 0.0001**

**Table 5 diagnostics-11-00223-t005:** Comparison of Fourier indices of the intermediately affected, affected, and unaffected eyes.

		Study Group vs. Controls Fourier Indices		
	Mean ± sd; (95% CI) *p*			
FECD Grade		Unaffected *n* = 70	Intermediately Affected (Grades 1–3) *n* = 22	Affected (Grades 4–6) *n* = 48
3 mm k Asymmetry		0.28 ± 0.14 (0.24–0.31)	0.49 ± 0.09 0.45–0.53 ***p* < 0.0001**	0.86 ± 0.26 0.78–0.95 ***p* < 0.0001**
3 mm k Higher Order		0.14 ± 0.02 0.13–0.15	0.18 ± 0.02 0.18–0.19 ***p* < 0.05**	0.19 ± 0.04 0.17–0.23 ***p* < 0.0001**
6 mm k Asymmetry		0.36 ± 0.19 0.31–0.40	0.66 ± 0.04 0.64–0.68 *p* < 0.0001	1.05 ± 0.29 0.96–1.14 ***p* < 0.0001**
6 mm k Higher Order		0.15 ± 0.03 0.15–0.16	0.23 ± 0.09 0.18–0.27 ***p* < 0.0001**	0.24 ± 0.07 0.22–0.24 ***p* < 0.0001**
3 mm p Spherical		−6.15 ± 0.2 −6.19 to −6.10	−6.01 ± 0.21 −6.11 to −5.93 *p* = 0.11	−5.76 ± 0.27 −5.84 to −5.84 ***p* < 0.0001**
3 mm p Asymmetry		0.04 ± 0.02 0.03–0.05	0.2 ± 0.12 0.14–0.25 ***p* < 0.0001**	0.32 ± 0.13 0.32–0.28 ***p* < 0.0001**
3 mm p Higher Order		0.02 ± 0.01 0.01–0.02	0.04 ± 0.01 0.03–0.04 ***p* < 0.0001**	0.06 ± 0.02 0.05–0.07 ***p* < 0.0001**
6 mm p Spherical		−6.13 ± 0.19 −6.19 to −6.10	−5.99 ± 0.23 −6.09 to −5.92 *p* = 0.09	−5.78 ± 0.26 −5.86 to −5.71 ***p* < 0.0001**
6 mm p Asymmetry		0.05 ± 0.03 0.04–0.06	0.22 ± 0.13 0.16–0.28 ***p* < 0.0001**	0.35 ± 0.1 0.33–0.39 ***p* < 0.0001**
6 mm p Higher Order		0.03 ± 0.01 0.02–0.03	0.04 ± 0.01 0.04–0.05 ***p* = 0.0001**	0.06 ± 0.02 0.05–0.07 ***p* < 0.0001**

## Data Availability

The data presented in this study are available in the article.

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
