# Peer review of "Corneal Analysis with Swept Source Optical Coherence Tomography in Patients with Coexisting Cataract and Fuchs Endothelial Corneal Dystrophy"

_diagnostics, 2021, doi:10.3390/diagnostics11020223_

Round 1

Reviewer 1 Report

The authors focused on defining the characteristic features of keratometry, pachymetry, elevation maps as well as Fourier indices based on swept source optical coherence tomography between normal and FECD eyes with coexisting cataract at different grades of disease severity. And they found that Corneal characteristic of FECD eyes independent of the significant corneal edema definitely should be considered during preoperative assessment in patients with coexisting cataract.

Major concerns

#1 Reduced FECD introduction info and expand SSOCT info.

#2 Authors must include chord mu and alfa data.

#3 CCT explanation is not solid, rewrite.

#4 Ocular surface and tear film measurement were not taking into account previous to the study

#5 Very small sample size for these results.

Minor concerns

#6 L30 Expand WHO

#7 L32 Why only white American data?

#8 L55-56 Provide an actual reference.

#9 L169-173 Expand all abbreviations.

#10 L182 Rewrite Table 1 in order to clarified data or describe in written form.

#11 L189 Cite formulas but do not include.

#12 Simplify Fourier analysis section.

#13 Reduce results section description.

#14 L385 Why only ray tracing, there are other options.

#15 L459 Provide a practical and day life explanation for this paragraph.

#16 Describe future research lines.

#17 Conclusion is weak, rewrite.

#18 Update reference 9, 24, 29 and 38

Reviewer 2 Report

This is the review for the manuscript "Tomography in patients with coexisting cataract and FUCHS endothelial corneal dystrophy" submitted by Anna Nowinska and co-workers. They used the CASIA2 device to study the corneal tomographic differences between cataract patients with and without FUCHS endothelia corneal dystrophy and found significant differences in posterior corneal parameters and anterior corneal  Fourier Indices. The results suggest that (1) posterior cornea should be considered for cataract surgery planning with FUCHS endothelial corneal dystrophy, (2) Corneal features might be useful for FUCHS screening in the future.

The manuscript is well written and the analysis appear sound. I have some minor comments:

1) You used both eyes of your patients. In normal patients right and left eyes show a symmetry, which might lead to an overestimation of the statistical significance. Can you comment how you adressed this?

2) Abbreviations in the abstract need to be introduced.

3) "The lack of agreement between anterior and posterior elevation..." (line 19, abstract). Please find a different wording for "a significant island of the negative values"

4) In the methods section (page 4, line 164) "Images not accepted by any of the ovserves were excluded from the study" Please add the acceptance criteria.

5) Table1: could you indicate in the table whether the differences between groups is significant/insignificant?

6) Formula 3: d needs to be in m and the 10^6 should be removed from the formula.

7) page 5, line 210: "20µm steps, green representing a point on the Best Fit..." I don't understand this sentence. What is green? Please rephrase.

Author Response

Dear Reviewer,

Thank you very much for valuable comments and clues. We carefully revised our manuscript and included all reviewers’ recommendations. We appreciate your comments and clues.

Answers and comments to the Reviewer 2:

The language English style was characterized as minor errors, so I decided to send the manuscript for English style editing and correction of grammar and spelling errors.

  • You used both eyes of your patients. In normal patients right and left eyes show a symmetry, which might lead to an overestimation of the statistical significance. Can you comment how you addressed this?

Thank you for your comment. I appreciate it. There are two main reasons, why we included both eyes in the study.

First - FECD usually develops asymmetrically, meaning that each eye of one individual is at different stage of severity level. Similar concept was brought by other researchers studying corneal characteristics in FECD, for example: Patel at al. (96 eyes of 56 patients); Wacker et al. 100 normal eyes of 54 controls, 32 eyes of 18 subjects, 43 eyes of 27 subjects and 37 eyes of 26 subjects at different severity level.

Patel, S.V.; Hodge, D.O.; Treichel, E.J.; Spiegel, M.R.; Baratz, K.H. Predicting the Prognosis of Fuchs Endothelial Corneal Dystrophy by Using Scheimpflug Tomography. Ophthalmology 2020, 127, 315-323

Wacker, K.; McLaren, J.W.; Patel, S.V. Directional Posterior Corneal Profile Changes in Fuchs' Endothelial Corneal Dystrophy. Invest Ophthalmol Vis Sci 2015, 56, 5904-5911.

Second reason is that we studied mirror left-right eye symmetry based on the anterior and posterior elevation maps. The right-left mirror symmetry is one of the defined features of the healthy corneas, observed in the process of optimizing the design of biosynthetic cornea. 72% of intermediately affected and 100% of affected FECD eyes lacked the symmetry in our study, mainly due to the presence of the posterior area of negative values in the elevation due to FECD. This observation was brought and discussed when I presented the preliminary results of this research during ARVO annual meeting in 2018.

  • Abbreviations in the abstract need to be introduced.

Thank you for the comment. All abbreviations in the abstract and the manuscript text were carefully checked and were explained.

  • "The lack of agreement between anterior and posterior elevation..." (line 19, abstract). Please find a different wording for "a significant island of the negative values"

Thank you for the comment. I changed the word “island” to “area”. Not only in the abstract, but in the whole manuscript text. I either changed it to “area” or used a term “focal posterior corneal surface depression”. I highlighted the changes to be easily spotted in the manuscript.

  • In the methods section (page 4, line 164) "Images not accepted by any of the ovserves were excluded from the study" Please add the acceptance criteria.

Thank you for your comment. We changed and highlighted the appropriate part of the methods.  [L159-172].

“The quality check of the scans were performed at two steps: automatically and manually. The SS OCT device instrument is equipped with an auto-alignment function and an auto-shot function that automatically initiated measurement when the subject’s eyes were within the proper range. Only measurements with a quality statement QS = ‘OK’ were accepted ad further analysed. QS is an index to show the reliability of the measured data calculated on following parameters: offset degree of corneal top is less than 0.86 mm (offset XY); area to be analysed is covered more than 93% within the front back corneal area of a 6 mm diameter (analysed area); contrast of the cornea is over 96.5% (valid data). When the measurement was completed, the preview screen with 16 radial OCT images with a green trace line on the corneal surfaces and crystalline lens was displayed and additionally the correctness of the analysis was confirmed by two independent observers (A.N., E.CH-T). Images, not accepted by any of the observers based on the alignment of the green trace line within the preview screen were excluded from the study”.

  • Table1: could you indicate in the table whether the differences between groups is significant/insignificant?

Thank you for the comment. We indicated the statistical differences by marking results characterized by p< 0,05 with “*”. The significant difference was noted and underlined for the value of BCVA of patients with 4-6 grades of severity (affected). The three groups included in the statistical analysis (controls, intermediately affected, grade:1-3 and affected, grade 4-6) were not significantly different in terms of age, sex and cataract severity grading and this was also underlined in the results part [L262-264]. According to the comments of the second reviewer, the Table 1 was simplified and reorganized. The new version of the Table was introduced to the revised version of the manuscript.

  • Formula 3: d needs to be in m and the 10^6 should be removed from the formula.

Thank you for the comment. I really appreciate this. We delivered the data from the producer description included in CORNEA/ANTERIOR Segment OCT CASIA 2, Tomey Corporation published on 1 Dec 2016. However, the second reviewer insisted on removing the formulas, so I removed them from the manuscript text adding only appropriate citations.

  • page 5, line 210: "20µm steps, green representing a point on the Best Fit..." I don't understand this sentence. What is green? Please rephrase.

Thank you. I agree with the comment. I corrected the sentence to avoid misunderstandings [L218-220].

“Color coded map with 20 µm steps are automatically implemented by the SS OCT software. In this map, the green color is representing a point on the Best Fit Sphere (BFS)”.

Round 2

Reviewer 1 Report

Comments solved